Blockchain and smart contract for IoT enabled smart agriculture

Pranto Tahmid Hasan
Noman Abdulla All
Mahmud Atik
Haque AKM Bahalul bahalul.haque@northsouth.edu
Electrical and Computer Engineering, North South University , Dhaka , Bangladesh
Taylor Ian
Electronic publication date: 2021 Mar 31
Publication date: 2021
Volume: 7
Electronic Location ID: e407
Received 2020 Nov 10; Accepted 2021 Feb 2
Copyright: © 2021 Pranto et al.
Copyright year: 2021
Copyright holder: Pranto et al.
License: This is an open access article distributed under the terms of the Creative Commons Attribution License, which permits unrestricted use, distribution, reproduction and adaptation in any medium and for any purpose provided that it is properly attributed. For attribution, the original author(s), title, publication source (PeerJ Computer Science) and either DOI or URL of the article must be cited.
License URL: https://creativecommons.org/licenses/by/4.0/

Keywords: Blockchain, Smart contract, IoT, Agriculture, Supply chain, Automation, Agricultural process, Transparent, Secure, Traceability

Funding: The authors received no funding for this work.

==============================
The agricultural sector is still lagging behind from all other sectors in terms of using the newest technologies. For production, the latest machines are being introduced and adopted. However, pre-harvest and post-harvest processing are still done by following traditional methodologies while tracing, storing, and publishing agricultural data. As a result, farmers are not getting deserved payment, consumers are not getting enough information before buying their product, and intermediate person/processors are increasing retail prices. Using blockchain, smart contracts, and IoT devices, we can fully automate the process while establishing absolute trust among all these parties. In this research, we explored the different aspects of using blockchain and smart contracts with the integration of IoT devices in pre-harvesting and post-harvesting segments of agriculture. We proposed a system that uses blockchain as the backbone while IoT devices collect data from the field level, and smart contracts regulate the interaction among all these contributing parties. The system implementation has been shown in diagrams and with proper explanations. Gas costs of every operation have also been attached for a better understanding of the costs. We also analyzed the system in terms of challenges and advantages. The overall impact of this research was to show the immutable, available, transparent, and robustly secure characteristics of blockchain in the field of agriculture while also emphasizing the vigorous mechanism that the collaboration of blockchain, smart contract, and IoT presents.

Introduction

Steady food supply across the world is solely dependent on agricultural activities around the world. The whole process of cultivation involves a lot of direct and indirect actors. Farmers, agricultural product sellers, and manufacturers are directly involved with the framework of cultivation. Furthermore, indirect actors are people who are depending on the production, e.g., the people who buy them to eat or use them as raw material to produce other variations of food or product. The process of agriculture is divided into two segments, the pre-harvest, and the post-harvest segment. The pre-harvest segment is basically the cultivation process, and post-harvest is composed of distribution and open market consumption of agricultural goods. However, the process gets initiated when seeds are brought to the storage. Seed storage has an enormous effect on seed quality, and seed quality has a direct impact on production. So, it is vital that seed storages are bought under monitoring. Data collection and regular interpretation of those data is a must to ensure a quality maintaining seed storage. Data should also be collected from cultivation fields for finding further insights. Distribution and market prices need proper monitoring. Ensuring the availability of traceable data for consumers is also necessary. Blockchain has already proven its capability in terms of safety where smart contracts bring automation, remove intermediate actors, and ensure proper regulation over the process. We can improve the overall agricultural system, which is also the primary motivation for this research.

Agricultural process in the pre-harvest segment includes processing the soil for seeding by sowing and watering, then adding fertilizers and composts, and the rest of the process involves irrigation and constant care-taking while expecting a fair amount of resulting crop to be harvested. However, it is not the case every time that a farmer ends up with the expected quantity of production even after following the appropriate orders and method of cultivation. Several factors might be involved in not having an expected amount of production, for instance, soil quality, low-grade seed and fertilizers, sudden drought or flood. Whichever among these reasons is causing low production of crops or sometimes completely demolishing the production, the farmers are always the one to be hit hard and suffer. Farmers are always in a constant state of risk starting from the very beginning of cultivation until he sells his product. On the other hand, according to a survey done by the US Bureau of Labor Statistics (BLS), farmers were the lowest-paid workers among several other worker categories (Fayer, 2014). The price in retail shops is much higher, sometimes twice or thrice, than the price sold by the farmers although the heart of the agricultural structure is the farmers, who keep the wheel of agriculture running.

Many farmers become disrupted, and they are not specialized in any other sector, which might have helped them for a swift profession shift. This condition of farmers creates depression, anxiety, and helplessness, often resulting in a very frustrating incident like suicide. A scoping review by Hagen et al. (2019) shows that approximately 225 million farmers in a year suffer from mental illness worldwide, where stress was most frequently found among farmers with a factor of 41.9%, followed by suicide with a factor of 33.1%. Nicole et al. (2020) showed that the suicide rate among farmers was increased to 37% between 2011 and 2017 in New Zealand.

The post-harvest part is significantly associated with the open market business. Agribusinesses are mainly targeted towards the consumers. The consumers around the world demand total identification and verification of agriproducts for better judgments of the products they buy from retail shops. The sudden growth of food-related hazards, diverse location of cultivation, and genetically modified organisms (GMO) have created a sense of awareness for which the consumer community nowadays is more conscious about having substantial evidence of the agriproducts being safe and nutritious (Opara, 2003). On the other hand, the pricing of the product varies on different layers of distribution, which needs more central control. There is a big question of which actor should be given what price in the range varying from farmers to retail shops.

For ensuring better monitoring of the market price, it is vital to trace the agriproduct starting from the inception point of seed storage, passing through the whole process of cultivation, and finally to reach the hand of the consumers. Agriproducts that the consumer buys should have concrete verifiable data, and the data will not only increase transparency but can also be used to monitor and manipulate the system. For instance, the quality of seed and the optimum environment for seed storage have an impact on food production. According to Kumar & Kalita (2017), approximately 50–60% of cereal grain can be lost due to the low maintenance of seeds. Pradhan & Badola (2012) showed how different storage conditions and storage period affects the seed germination process of Swertia chirayita, a Himalayan plant. Contrarily, they found that 4° is the optimum temperature for storing this seed for a long time (Pradhan & Badola, 2012). So, there is a clear need for using technology that monitors the environment of storages that stores the seed and other agricultural products (e.g., fertilizer, pesticides).

Solving these problems in the agricultural system is very important in order to keep continuous food supply running without facing the global food shortage. Technology like blockchain (Nakamoto, 2008), along with Smart contracts (Szabo, 1997) integrated with Internet of Things (IoT) devices, can solve these problems by providing a distributed network of connected sellers and buyers. An organization or suitable authority will monitor all the relative information in the system and set the prices of agricultural goods and services. IoT devices will monitor the quality of seed and fertilizer and trigger events in the blockchain network if anything goes wrong along the process of farming. The smart contract will be deployed inside the blockchain network so that it cannot be changed or tampered by anyone. The business terms and conditions applied in agricultural transactions become solid and immutable.

The main contribution of this paper is as follows.Ensure traceability of agricultural products from root to retail.

Ensure temper proof data acquisition from storage, field level, and while distributing.

Remove intermediate processors and controlling market prices.

Gain more control over the process by Smart Contracts.

Automation of the process while removing intermediate third-party processors.

And lastly, secure environment implementation with blockchain for better security of valuable data.

The paper is arranged in six sections and several sub-sections. The rest of the paper is arranged in the following manner. “Literature Review” contains the literature review where we have discussed blockchain, smart contracts, IoT devices while focusing on using these technologies to improve processes and enhance security. “PRoposed Blockchain-based Model” contains an extensive overview and the design of our system. How blockchain and the other supporting tools like MQTT network protocol and IoT devices interact with the system have been described in this section. “Implementation and Testing” includes the implementation and testing where we have shown the implementation details along with the algorithms that we have used. The system test outputs have been displayed in this section. In “Analysis”, we have analyzed our model in terms of advantages and disadvantages. We also showed a gas cost analysis of the operations. Finally, the conclusion section (Conclusion) contains the outcomes and future research directives.

Literature review

Blockchain and smart contracts have already proven its capability for process development that requires transparency and concrete evidence-based record keeping. While blockchain establishes a sturdy trust, the smart contract makes sure that the necessary logics and rules are implemented automatically without human manipulation and intervention from a third-party. On the other hand, IoT devices provide excellent technical support when it comes to monitoring a process by collecting and sending data over the network. This section will be doing a comprehensive literature review of blockchain, smart contract, and their ability in terms of tracing, monitoring, and overall development of real-world scenarios.

Blockchain

Blockchain is a disruptive technology that has been entitled as the “the most important technology since the internet itself” by an influential Silicon Valley Capitalist Marc Andreessen (Crosby et al., 2016). This extremely robust technology is well described by the naming of itself. Blockchain is nothing but a chain of connected and verified blocks where each block contains some transaction data (generally represented as Merkle tree) and the cryptographically hashed address of the previous block along with the timestamp (Zheng et al., 2017). The first block is named the genesis block. The structure of the blockchain is shown in Fig. 1.

Figure 1 Structure of blockchain.

The most popular and well-known usage of blockchain goes by the name of bitcoin. An unknown quantity of people under the name Satoshi Nakamoto first published the whitepaper of bitcoin on 31st October 2008 (Nakamoto, 2008), and the first business implementation was in action in the following year. Timestamping the digital payments by the use of cryptographic hash is the paramount convenience of blockchain. On the other hand, blockchain solves some significant problems in digital payments. Double spending is the idea of spending the same digital payment token more than once. The dilemma of double spending is created by making false claims and distorting information to create a disguise (Hoepman, 2010). However, blockchain strictly handles this situation and does not allow any distortion in the data to be stored in the blockchain. This tamperproof mechanism is implemented through a distributed consensus algorithm (Nakamoto, 2008). Every transaction is validated by most of the users connected in that blockchain network. So falsifying information is nearly impossible in blockchain technology. After the first block (The genesis block), each and every block is added through distributed consensus along with the cryptographic footprint and timestamp (Nofer et al., 2017). This information is updated in every user node connected to the blockchain. So, double spending becomes infeasible with the usage of blockchain technology. Transaction in the blockchain is shown in Fig. 2.

Figure 2 Transaction in blockchain.

Blockchain has specific characteristics which are not only applicable for securing digital payments but also suitable for abolishing third-party based trust models in businesses and organizations (Nofer et al., 2017). While using blockchain, third-party as a financial processor is no longer required. The process is automated through smart contracts and tamperproofed by the blockchain itself. Each transaction history is logged into the blockchain and accessible at any time in the future (Nofer et al., 2017). The research community is working on applying this technology for real-world problems that require more sophistication in the transaction and payment process. Blockchain seems to have great potential in the future of digital transactions for use cases like insurance (Raikwar et al., 2018; Lamberti et al., 2018) and banking (Eyal, 2017; Peters & Panayi, 2016). Blending real-world online payment systems with blockchain often bring about the problem of scalability in terms of the number of transactions and computational power. Studies now prove that implementing blockchain-based systems with a feasible number of payments and computational power can be done (Zhang et al., 2018) while also establishing scalability by reducing block weight and ledger size in global peers (Biswas et al., 2019). Another problem in digital platforms is the gap between payment and receiving goods or services in exchange, which is known as payment lead time. Chang, Chen & Lu (2019) showed a process of re-engineering the supply chain using the blockchain technology while reducing payment lead time in the digital payment system. Thus, blockchain presents splendid capability for online transactions in terms of security and transparency while eliminating the existing complications in digital payment systems.

Contrarily, blockchain shows immense prospect in the sectors that do not necessarily involve digital payments but uses the characteristics of blockchain, for instance, distinctions of types(public, private, permissioned), access control (centralized, decentralized), persistency, validity, identity control, transparency control(closeness or openness) and superior security (Zheng et al., 2018). A practical implementation of blockchain-based agri-food traceability has been shown by Caro et al. (2018). They showed how transparent, fault-tolerant, immutable, and audible tracing could be done using blockchain and IoT devices. Manupati et al. (2020) showed how the use of a distributed ledger system of blockchain reduces total cost as well as carbon emission in a multi-echelon supply chain. Blockchain has been used in storing and processing information. File security in the blockchain is far better than any other existing cloud storage system, while the transmission delay is significantly lower (Li, Wu & Chen, 2018). File loss rate in currently available cloud storage architectures can be up to 100%, where it is nearly 0% in the blockchain (Li, Wu & Chen, 2018). Decentralized storage (blockchain) of Interplanetary file system to store and share industrial spare parts data has been implemented by Hasan et al. (2020). Blockchain is one of the most secure ways of dealing with electronic transactions. However, the usage of blockchain in real-world systems that deal with raw data is shown by many researchers. Substantial security in the mobile cloud data of Electronic Patient Record Systems (EPRs) is shown by Nguyen et al. (2019) using blockchain technology. Implementing blockchain in real-world use cases shows excellent future potential, as discussed in this section.

The essential idea of blockchain was to create a robust online transaction system. Blockchain’s fundamental aspects have presented so many sturdy prospects of excellent technical support over distributed systems in terms of security and transparency. However, blockchain does not have an integrated system for automatically processing the data in a distributed system. We can achieve this by integrating a smart contract with blockchain.

Smart contract

With the help of blockchain, we can assuredly discard intermediaries, but the promises and trust boundaries between the contributing parties frequently present the requirement of what is called a smart contract (Macrinici, Cartofeanu & Gao, 2018). Like traditional contracts, the smart contract is also a collection of organizational terms and conditions that regulate the trust between the parties involved within the scope of that contract. The only difference is that a smart contract is coded with a programing language. The rules, terms, and conditions are implemented via controlled coding, reflecting the exact agreement approved by all the parties (Szabo, 1997).

The idea of smart contracts was there from the 1980s, but the only thing it lacked was the removal of intermediaries. Nick Szabo first published the whitepaper of smart contracts in 1996. According to Szabo,

“The basic idea of smart contracts is that many kinds of contractual clauses (such as liens, bonding, delineation of property rights, etc.) can be embedded in the hardware and software we deal with, in such a way as to make breach of contract expensive (if desired, sometimes prohibitively so) for the breacher.” (Szabo, 1996)

So, the main objective of smart contract was to embed the contractual clauses inside a combination of hardware and software so that breaching becomes difficult, and the cost of a contractual breach becomes prohibitive, ultimately increasing the safety of contracts and decreasing the possibility of an attack. The idea of smart contracts and the real-life implementation of it was popularized in 2016 by Ethereum blockchain. Ethereum is a decentralized Turing-complete blockchain which has integrated tools and environment for implementing smart contracts (Tikhomirov, 2017). The gap that Ethereum has filled has made Nick Szabo’s statement possible in a real-life scenario. Szabo described smart contracts as a “contractual breach cost increasing mechanism” that reflects the actual contract. As Ethereum itself is a blockchain, storing the contract inside the blockchain makes it tremendously challenging to break in and tamper with the contract. A typical smart contract building and its execution steps are shown in Fig. 3.

Figure 3 Steps of building a typical smart contract.

A smart contract, in other words, is the automation mechanism of blockchain technology. This very idea of storing the contract inside the blockchain has opened the door to several other implementation possibilities of blockchain in real-world problems. Chang, Chen & Lu (2019) uses smart contracts as their core of system design to automate the processes involved within the system. The automation includes the real-time tracing of products in a supply chain and overall control over all the steps involved. Hasan et al. (2020) integrates smart contracts in their industrial spare parts traceability research work to implement the necessary function, modifiers, and events inside their proposed system, which mainly controls the logical flows that automate the process by using the smart contract.

The essential need for trust establishment between parties can be achieved through smart contracts by securing the contract inside the blockchain (Bader et al., 2019) has proposed an implementation of a smart contract-based car insurance ecosystem named CAIPY in which the smart contracts implement step by step process of the insurance policy as well as interacts with the tamperproof IoT devices for keeping information about the car condition. Intellectual rights management can be done using smart contracts. Zhao & O’Mahony (2018) show the implementation of a music copyright management system named BMCProtector that uses blockchain and smart contracts. The smart contract in their system implements the necessary function starting from music creation till royalty distribution. As the contract has been distributed inside blockchain, it is almost impossible to break in and change it, which justifies absolute security while deploying a smart contract inside the blockchain environment (Bader et al., 2019; Zhao & O’Mahony, 2018).

Coding a smart contract often comes across some common terms like attribute, function, event, and modifier. These terms are described below.Attributes: Attributes are the variables which holds the values in the memory. Solidity programing language allows attributes for different primitive data types (int, char, string, double), mapping, address, and enumerators.

Functions: Function represents a mechanism or a task inside the system. Once a function is called, the task that has been written inside the function body will be executed.

Modifiers: Modifiers represent the access power of the actors or components. While the contract owner has supreme modification power, other actors or components can be given some specific modification or access power by the contract.

Events: Event aids the purpose of storing anything in the transaction log of blockchain. When an event takes place or is emitted, the data passed on the event as an argument gets logged in the transaction logs of the blockchain. By this mechanism, historical data of the system is stored which can be retrieved later. This event trigger mechanism makes the system auditable.

From the above discussion about the usage of blockchain and smart contract in various fields and use cases, it is quite clear that the combination of blockchain and smart contract results in an ingenious, automated, and highly secured system. While blockchain provides a robust platform to store and track the system’s data, a smart contract implements the business logic and controls the access distribution. In our project, several necessary data come from outside the system. We intend to use IoT sensors for this purpose.

Blockchain, IoT, and smart contract for traceability and process development

A smart contract, united with blockchain and integration of IoT devices, has been proven as a smart, secure, and reliable course of action to trace and monitor over processes and operations. Without these technologies, the current scenario lacks a well-proved medium for tracing or monitoring systems while contributing significantly to the quality control and development of the process with robust security. Smart contracts have opened a door towards this development with all the necessary digital means of support intending to automate the tracing process and establish a trustless rigid contract between parties involved. One of the major drawbacks in the traditional process management and traceability is that the data can be manipulated at any stage, and there is no security that the business rules will be strictly followed in the future.

To secure data inside the blockchain, accumulation must be done in the first place. IoT devices have already been proven excellent for monitoring and collecting data with low power and minimum cost (Pavithra & Balakrishnan, 2015; Tapashetti, Vegiraju & Ogunfunmi, 2016; Baranwal, Nitika & Pateriya, 2016). Effective and cost-friendly home automation using IoT devices has been implemented by Pavithra & Balakrishnan (2015). A low-cost air quality monitor system that collects data from open-air and analyzes the data is an adequate use of IoT devices which has been shown by Tapashetti, Vegiraju & Ogunfunmi (2016). Although IoT-based devices have many issues regarding information security (Miloslavskaya & Tolstoy, 2019; Vashi et al., 2017), using blockchain as the secured backbone solves these issues (Mohanty et al., 2020).

Traceability of a supply chain or production using IoT, blockchain, and smart contracts can resolve many business processes to make them smooth and save our valuable time. The research community also has much interest in integrating those technologies to solve some life-associated problems. Kim et al. (2018) showed how to design a food traceability system using IoT, integrate with blockchain, and smart contract. Lin et al. (2018) showed how blockchain and IoT based food traceability models could be introduced in the smart agriculture ecosystem. A case study conducted by Lucena et al. (2018) shows that the grain quality assurance tracking using a real scenario with a blockchain-based business network, which will append the valuation around 15% of GM-free soy in the grain exporter business network in Brazil. Tracing supply chain or a process contributes towards the quality of management of these processes (Chen et al., 2017) proposed a four-layered architecture to improve supply chain management’s quality by adopting blockchain, where IoT devices (GPS & Sensors) are applied in the very initial layer. All these researches indicate a positive future of blockchain in the management of supply chains and processes.

The most crucial feature of blockchain is immutability, which can safeguard these accumulated data as no data can be manipulated inside the blockchain without altering the whole sequence or history (Galvez, Mejuto & Simal-Gandara, 2018; Casino, Dasaklis & Patsakis, 2019). Moreover, real-time tracing of any process can be done using blockchain (Tian, 2017). Complex and sophisticated industrial processes can be automated and tracked by real-time data using blockchain technology (Westerkamp, Victor & Kupper, 2018). Modern-day enterprises require symmetric information flow along the way, proper regulation, and availability of legacy information. To date, blockchain provides the perfect solution to these requirements (Kim et al., 2019). The advantages of blockchain and smart contracts are being applied to sectors like precast construction (Wang et al., 2020), medical services (Chen et al., 2018), and transportation (Humayun et al., 2020) sector. Wang et al. (2020) uses blockchain to improve the current scenario in precast construction where low fragmentation and scarcity of real-time data becomes a problem. They automated the data sharing process and ensured information traceability and transparency in precast construction. The research community has shown the profound prospects of blockchain ranging from medical record keeping with absolute security (Chen et al., 2018) to intelligent logistic support for transportation systems (Humayun et al., 2020).

Blockchain possesses excellent potential in terms of transparency in food-related processes (Kamilaris, Fonts & Prenafeta-Boldv, 2019). Better traceability of a food supply chain using blockchain has been shown by Wang et al. (2019). Their system implements a response mechanism for various events for an improved, validated, and guaranteed transaction. In a survey paper, Lin et al. (2020) showed how blockchain and IoT-based systems contribute to food safety, food security, food quality monitoring, and control to support small-scale farmers. Legacy information is vital for making decisions about a product as food can be categorized and sometimes prohibited for several reasons. Tan, Gligor & Ngah (2020) proposed a traceability system for halal food chains to ensure that the food processing steps follow strict measurements so that the food remains halal from frame to fork.

Quality control and quality improvement is a must for sectors like agriculture. The existence of humankind is significantly related to the sustainable food supply. The use of technologies like IoT, blockchain, and smart contracts will facilitate the improvement of the current scenario in agriculture. These improvements are inevitable to have a traceable, rigid, and secure agricultural process while gaining more control over the whole process.

Proposed blockchain-based model

System overview

We propose the usage of IoT enabled smart actors for a better mechanism of necessary data flow across the system. IoT devices will be used to monitor the quality and condition of the products stored in the large storehouses. They will also monitor and send data about the pricing of agricultural goods and services for both pre-harvesting and post-harvesting periods. IoT devices will also provide information during the cultivation process. Blockchain is used to safely store this monitored data while a Smart Contract will be used to automate the process, trigger events, and set the necessary implementation of terms and conditions for all the parties. The general system overview is shown in Fig. 4.

Figure 4 Overall system overview.

As exhibited in Fig. 4, the main actors in the system are the storage warehouses, supply shops, producers, distributors, wholesalers, and retailers. At the very beginning, the administrative body deploys the contract within blockchain network, which is marked as step 1. Then, in step 2, topic-subscription mechanism is implemented along the whole chain of storage and distribution. This enables the IoT devices to get connected to the server. In the following steps starting from step 3 and ending at step 8, data is being accumulated from IoT devices and is stored to the MQTT server while some sophisticated data is stored in the blockchain. Periodical checks are done by the smart contracts for security, traceability, and quality maintenance purpose. The system actors are connected to an Message Queue Telemetry Transport (MQTT) (Hunkeler, Truong & Stanford-Clark 2008) cloud storage by creating a topic-based subscription-publish system. The standard HTTP connection requires the establishment of a connection each time a request is made to the server, but MQTT does not require that (Wukkadada et al., 2018). As it is a topic subscription-based system, it only needs one valid subscription to the topic created in the server. MQTT performs faster than HyperText Transfer Protocol (HTTP) for IoT sensor data accumulation, and it is also lightweight compared to standard HTTP (Wukkadada et al., 2018). Our system will not store each and every data to the blockchain as there is a storage limitation in blockchain. Data, in general, will be stored and made available via MQTT servers, but sophisticated data will only be dealt with blockchain. MQTT aids the purpose of storing, sharing, and publishing data that enables aggregation of data among all the parties within our system.

System design

The system consists of actors like seed storages, supply shops, producers, distributors, wholesalers, and retailers. Another actor here is the contract deployer. Several components will achieve the interaction mechanism between the actors and the system. The role of each actor and components is demonstrated in the sections below.

Actors

The heterogeneity of several actors imposes a common threat of a reliable, immutable, and verifiable system. Our proposed model connects these actors via multiple technological resources. The characteristics of the actors are discussed below.Contract owner: The contract owner has superior power over the system. The owner deploys the contract on the system and monitors if the regulation has been implemented correctly or not.

Seed Storage: The storages primarily store seed and other agricultural products. Sensitive (sun exposure, temperature) seeds and agricultural products are stored on a large or medium scale in storages for a comparatively long period of time.

Supply Shops: The supply shops collect a large amount of seed, fertilizer, and all other necessary agricultural products and sell them to the producers. The storage period is shorter for these supply shops.

Producers: The farmers are the primary level producer. They execute all the tasks related to planting and harvesting crops.

Distributor: Distributors are responsible for navigating the crops safely from one place to another.

Wholesaler: The wholesalers buy crops and agriproducts on a large scale and sell them to retailers.

Retailer: Retailers buy the crops and products from the wholesalers and sell them in open markets directly to the consumers on a small scale.

Consumer: Consumers are the mass people who depend on agricultural products and play an essential role in the system by continually creating demand.

The main endeavor is to build a system where these actors interact in such a way that contributes toward the overall traceability of these products.

Components

Several components will be used to implement the system requirements like immutability, availability, security, removing intermediate third-parties and automation. Blockchain, smart contract, IoT enabled environment, MQTT server is the fundamental components of our system. The components are described below.

A. Blockchain

Blockchain abets the authenticity and reliability of the system. The data accumulated in our system is primarily stored in the MQTT server. MQTT lacks security in some of its steps, where it does not encrypt the data (Andy, Rahardjo & Hanindhito, 2017). As one of our main motives for this work is to increase the transparency of the trackable data of agricultural products, tamperproofing the historical data is a must. We use blockchain specifically for this sole purpose. The diagram below (Fig. 5) shows the system interaction with the blockchain.

Figure 5 System interaction with blockchain.

Several events are triggered during some essential steps of the agricultural process, and the data is logged into the transaction log of the blockchain. This data can never be changed or tampered with without breaking the chain of blocks. So, traceable data becomes secure with blockchain.

B. Smart contract

Smart contracts use a combination of attributes, functions, events, and modifiers to automate the processes and remove the dependency on intermediate third-parties. Where attributes represent storage variable, function represents the execution of a task, event represents the occurrence of selected statements, and modifier represents the authority of actors. Our system implements two smart contracts written in Solidity programing language and uses Ethereum based blockchain technology as the core apparatus of our work. The first contract is for storage, and the second contract is for distribution.

Figure 6 shows the attributes, functions, events, and modifiers that the storage smart contracts hold, possess and implement.

Figure 6 Smart contracts used for storage in the system.

The storage contract (Fig. 6) automates the data collection and event logging from the storage that we store in the blockchain. The contract can self-check for the data and compare it with the optimum values. Based on the result, the contract automatically logs events inside the blockchain, securing verifiable information for future buyers and quality measurements. The second contract is used to track the agriproducts after the production. One of the major problems that our system tries to solve is authenticity in trackability data of agriproducts and making it auditable to the general customers. The second contract can also be used to monitor the pricing. In every step from producer to customer, this contract will keep track of the dates along with the price and quantity sold.

Figure 7 shows the attributes, functions, events, and modifiers that the distribution smart contracts hold, possess and implement.

Figure 7 Smart contracts used during distribution.

C. IoT enabled environment

IoT devices, mainly low power sensors, will be used to update regular real-time data from the environment. The diagram below (Fig. 8) shows how the integrated IoT devices interact with the MQTT server and show how the published data can be conveyed or routed towards users.

Figure 8 IoT enabled environment interaction with the MQTT server.

D. MQTT network protocol

For fabricating a collaborative behavior among the sensors, the blockchain, and the actors, we suggest using an Message Queuing Telemetry Transport (MQTT) network protocol. MQTT is a network protocol that requires minimal bandwidth and consumes very low memory. IoT devices or the sensors will read data like temperature, humidity, and light exposure from the environment, and that data needs to be shared across the whole system eventually to the users. MQTT protocol provides such functionality that enables us to do that. MQTT provides a subscription and publishes based model. The system will have several topics on the MQTT server, and the clients will subscribe to specific topics for looking into the data and publishing data frequently to the server under that topic.

Implementation and testing

We used an Ethereum, blockchain-based environment to implement and test the mechanism of blockchain. For writing smart contracts, we selected Solidity. The Remix environment was used for creating and testing the core implementation of the proposed system. Building the whole system is not the purpose of our interest. However, an architecture directive approach is demonstrated throughout our paper. In this section, we shall discuss the implementation in detail and also show the testing results.

Implementation

The objective of our work is to show the applicability and opportunity of blockchain in the field of agriculture while ensuring traceability. The fundamental goal of this paper is to demonstrate how the use of blockchain and smart contracts can trace the agricultural products from field level production and continue tracing until the product reaches the consumer base. The system also monitors the pricing along the process. Figures 4, 5 and 7 shows clear interaction among the system components and actors. We considered the process into segments, the pre-harvesting period, and the post-harvest period. The working procedure in both segments is discussed below.

A. Pre-harvest

In the pre-harvest period, the system monitors the storage condition that directly impacts the quality of seed, that is, temperature, humidity, and light exposure. Smart contract can execute self-checks for these values and perform a violation trigger. Self-check follows the algorithm below.

Algorithm 1 Self check algorithms.

Result: Smart contract conducts a periodical self checks.	
Data: value read by IoT device.	
1 if caller == owner then	
2   if value > optimumValue then	
3     call violationTrigger() method with violation type and category ;	
4     return a string describing the high value violation ;	
5   else if value < optimumValue then	
6     call violationTrigger() method with violation type and category ;	
7     return a string describing the low value violation ;	
8   else	
9     call violationTrigger() method with violation type and category ;	
10    return a string describing that the value is optimum ;	
11  end	
12 else	
13  Do Nothing ;	
14 end	

Registered storage is equipped with IoT sensors that will provide these data to the system on a periodical basis that is essentially stored in the MQTT server. Through the smart contract, the system can auto-check these data using the temperatureSelfCheck(), hummiditySelfCheck() and lightExpoSelfCheck() method while comparing the observed value with pre-defined optimum values. These methods use another trigger method named violationTrigger() to emit events in blockchain so that any violation during storage is stored inside the blockchain along with the violation timestamp. Violation trigger follows the algorithm below.

Algorithm 2 Violation trigger algorithm.

Result: Smart contract emits a violation event.	
Data: ViolationType and ViolationCategory	
1 if caller == owner then	
2   if violation type == Temperature then	
3     if category == 1 then	
4       set the temperature condition Enum to Over ;	
5       set the violation type Enum to temperature ;	
6       emit TemperatureViolation event describing that the temperature is over the threshold;	
7     else if category == 0 then	
8       set the temperature condition Enum to Under ;	
9       set the violation type Enum to Temperature ;	
10      emit TemperatureViolation event describing that the temperature is under the threshold ;	
11    else if category == 2 then	
12      set the temperature condition Enum to Optimum;	
13      set the violation type Enum to Non;	
14      emit TemperatureViolation event describing that the temperature is optimum;	
15    else	
16      Do Nothing;	
17    end	
18  else if violation type == Humidity then	
19    follow statement 3-18 with associated values, parameters and conditions;	
20  else if violation type == Light Exposure then	
21    follow statement 3-18 with associated values, parameters and conditions;	
22  else	
23    Do Nothing;	
24 else	
25  Do Nothing;	

The violationTrigger() method is called with the type of violation (temperature, humidity or light exposure) and the category value with “1” being over, “0” being under, and “2” being optimum with respect to the pre-defined optimum value. According to the violation type, the enumerator value is set to Temperature, Humidity, or Light Exposure, and according to the value of category, events are logged into the blockchain accordingly.

B. Post-harvest

In the post-harvest, the system tracks all the necessary information and logs events in the blockchain. Information includes departure dates, prices along the way, and quantity in every layer that the product crosses till it gets to the retail. Tracking starts by invoking initiateDistribution() method. This method takes information like product id, name, price, and quantity. The method sets the date as the current timestamp of the block and sets the trace enumerator to the producer. Distribution starts by invoking the startDistribution() method, and the previous information gets updates accordingly while setting the trace enumerator to Distributor. The same mechanism is applied to the other layers of agriproduct supply. Figure 11 shows how every information is timestamped within blockchain.

Figure 11 Triggering a violation through self check of temperature condition (A) and example of initiating a distribution (B).

As we can see from Fig. 9, tracing data in our system starts from the production level. As farmers sell his product, the product information and the block timestamp are stored by emitting events in the blockchain so that these data remain secure and become verifiable. As the algorithm is the same in all the steps, only the producer level algorithm (Algorithm 3) is shown below to demonstrate the process.

Figure 9 Timestamping distribution data to blockchain.

Algorithm 3 Product trace initiation algorithm.

Result: Product distribution starts.	
1 Only producer account can start the distribution process.	
2 if caller == owner then	
3   Set the product CurrentTrace enumerator to producer ;	
4   Set initiation date to the current timestamp of the block ;	
5   Set name, id, price, quantity to the associated variables ;	
6   emit DistributionInitiate method with producer address and a message describing the product distribution starting ;	
7 else	
8   Do Nothing ;	
9 end	

The below algorithm (Algorithm 3) is applied to the other layers of distributed tracing. Only the relevant data and authority changes.

Testing

We provided the system architecture, and our implementation includes the smart contract interaction with Ethereum blockchain. Remix is a web-based IDE that provides Ethereum wallet as well as accounts loaded with dummy ether cryptocurrency and the environment to write smart contracts with the help of Solidity programing language. Remix also provides us the environment to run and deploy the contract inside the Ethereum blockchain, which matches our fundamental requirements and aims. In this section, we provide a demonstration of our contracts interacting with the blockchain, and we will also demonstrate the mechanism it follows.

In the post-harvesting period, seed information is inserted into the system and, with the help of IoT devices, corresponding autority or monitoring body can check for the current status of temperature, humidity, or light exposure periodically. Based on the self-checked data, the contract calls events to store the favorable condition or violation along with their type and condition. Where type is temperature, humidity, or light exposure and condition values can be indicated as below, over, or optimum (than the threshold). The figure below (Fig. 10) shows the successful entry of a seed (Fig. 10A) and demonstration of a self-check result (Fig. 10B).

Figure 10 Add seed to the system (A) and temperature self-check in the storage (B).

Using the addSeed() method, necessary seed information is stored inside the blockchain for a secure safeguard of legacy data. For demonstration purpose, an entry of a batch of potato seed has been inserted into the blockchain as shown in Fig. 10A. All the registered seed storages will insert seed data as soon as seeds are brought for storage purpose.

Several conditioning reason will be monitored using IoT devices. Our system implement checks for crucial factors like temperature, humidity and light exposure. All these monitored data will be stored in clouds but for establishing an absolute trust-based decision making, it is important to know that these data are not corrupted or altered or hampered in any way. So, the storage smart contract conducts periodical self-checks for this contributing factors. Self-check data is analyzed and detailed information is stored via events in blockchain. Figure 10B shows a demonstration of the contract executing a self-check for temperature.

The optimum temperature was set to 38. When the contract checks with value 40 (read by IoT devices), it finds a violation in temperature value and instantly emits the TemperatureViolation() event. The event gets logged in the transactional history along with the address(which storage) and violation information (what kind of violation).

The contract also triggers violation event whenever any violation is encountered by the system. Based on the violation type and category of violation, event are logged into the blockchain. Log data are retrievable and farmers will be able to see and make decision based on these historical data of any particular storage. Trigger testing is shown in Fig. 11A. As depicted in Fig. 11A, the violationTrigger() method successfully logged the violation of temperature attribute in the blockchain.

In the post-harvest segment, the system tracks every product by storing data in each and every step of distribution, starting from the production level to retail. These information is essential for the customers to know the origin and distribution of products before buying it. From producer to distributor then wholesaler and finally retailer, every step does a similar fashion of interaction with the system. The producer level demonstration is shown in the Fig. 11B.

As shown in Fig. 11B, using the initiateDistribution() method, the process of distribution has been started. The log shows all the data related to the agriproduct from producer level. Product id, date, price, and current trace is logged into the blockchain. All the other steps of distribution does the same thing with relevant data from the product’s current trace, finally, the product reaching to retail shops.

Analysis

Though transaction with blockchain has been popularized bitcoin, implementing it in a real-world use-case, with the integration of other technologies like IoT, still has several issues to solve and several purposes of the meeting (Casino, Dasaklis & Patsakis, 2019). The use of IoT in agricultural traceability is being encouraged by the research community (Lezoche et al., 2020; Hiromoto, Haney & Vakanski, 2017; Abdel-Basset, Manogaran & Mohamed, 2018; Ben-Daya, Hassini & Bahroun, 2019). Though our system tries to avoid dubious conditions by implementing smart regulatory contracts, there are still some flaws in the system. As blockchain is still in its inception stage, there’s an issue of optimizing it for real world systems (Casino, Dasaklis & Patsakis, 2019). Table 1 show a comparison analysis of our system with recent works in the field of agriculture using blockchain. The challenges and advantages represent the analysis of our system. A gas cost analysis has also been attached in Table 2.

Table 1 System comparison with existing methods.

Outcome	Tian (2017)	Caro et al. (2018)	Kim et al. (2018)	Lin et al. (2018)	Devi et al. (2019)	Salah et al. (2019)	Umamaheswari et al. (2019)	Kamble, Gunasekaran & Sharma (2020)	Bumblauskas et al. (2020)	Shahid et al. (2020)	Proposed System	
Provide system implementation	✗	✓	✗	✗	✓	✓	✗	✗	✓	✓	✓	
Traceability	✓	✓	✓	✓	✗	✓	✓	✓	✓	✓	✓	
Control over the system (Smart Contract)	✓	✓	✓	✗	✓	✓	✓	✓	✓	✓	✓	
Enable informed purchase decision ability of customers	✓	✓	✗	✗	✗	✗	✗	✗	✓	✗	✓	
Real-time data	✓	✗	✓	✓	✓	✗	✓	✗	✗	✗	✓	
Reduce fraud	✓	✗	✓	✓	✓	✓	✓	✗	✓	✓	✓	
Remove third party	✓	✓	✓	✓	✗	✓	✓	✓	✓	✓	✓	
Fair pricing	✗	✗	✗	✗	✗	✗	✓	✗	✗	✗	✓	

Table 2 Gas costs of different operations.

Task	Associated actor	Transaction cost	Execution cost	
addSeed()	Storage	168,402	144,314	
temperatureSelfCheck()	Storage	50,681	29,217	
hummiditySelfCheck()	Storage	50,812	29,348	
lightExpoSelfCheck()	Storage	35,503	14,039	
violationTrigger()	Storage	48,625	26,969	
initiateDistribution()	Producer	132,122	108,610	
startDistribution()	Distributor	106,442	84,786	
startWholesale()	Wholesaler	91,530	69,874	
retailSell()	Retailer	91,464	69,808	

A. Advantages

Authorization is never compromised. Only the person allowed by the smart contract can execute or invoke a task.

Monitored data has been secured and solidified in two ways. IoT devices automatically push data to the MQTT server and the blockchain periodically self-checks these data via smart contract and logs every event and violation.

Data security within the blockchain is robust. Several types of data from pre-harvest and post-harvest segments are stored in the blockchain so that they remain unchanged as any data saved into the blockchain can hardly be changed or tampered.

Authenticity has been made sure by the smart contracts. The smart contract automatically checks values in different segments and saves the result within the blockchain.

A central source of information is created by the use of blockchain, and everyone can check the necessary data at any time.

A smart contract removes the intermediate processor. In agriculture, there are so many intermediate person/factors which increases the market price drastically. However, a smart contract monitors all the prices and the flow of product from one level to another.

Every date, times, and product id are stored in the blockchain. So, authentic, traceable data ensures the consumer’s right to know about the product’s origin.

Central monitoring organizations can use this to control market prices and also to estimate supply-demand.

The processes are automated and robust.

There is no third-party support needed. Only the system alone can handle all the operations without any human interaction.

B. Challenges

If, in any step, someone does something by mistake, changing or updating data in the blockchain is impossible. Immutability indeed is one of the biggest strengths of blockchain, but unlike the traditional database systems, blockchain, due to its reliable security schemes, does not allow us to modify the data of a verified transaction.

Smart contracts are deployed to take control of the system and automate the steps. Once deployed inside the blockchain, it can never be changed.

The initial settings of the blockchain environment take a reasonable amount of money.

IoT devices are vulnerable to security issues.

Data acquisition is dependent on IoT devices. If devices get damaged, data collection or check is not possible.

C. Cost estimations

The table above (Table 2) show the gas costs of different operations within our system.

Conclusion

The objective of this work was to demonstrate the capability of blockchain in agricultural process development, and how the usage of blockchain can ensure data availability, ensure security, apply immutability so that there can be no manipulation of data, and ensure trust among producers and consumers. Consumers can, without any doubt, know the origin and distribution history of products. Farmers can know the history of seed storage, and the governing bodies can control the market based on these data. The whole system is bought under IoT surveillance, where blockchain provides absolute security of them. All codes related to the system have been made public in GitHub (https://github.com/Tahmid1406/Blockchain-Based-Agriculture) so that other enthusiasts can learn from it and use it for their purpose. The implementation is quite generalized and generic. The same implementation can be applied to trace, develop, and enhance any process. We also presented the gas costs for a better understanding of the costs to run the system. Our future goal is to deploy the system in a permissioned blockchain while ensuring that tracking becomes full-fledged. For tracking purpose, we intend to use barcode or QR code from storage to retail sale. We also intend to build a web application that meets our purpose.

Supplemental Information

Supplemental Information 1 Smart Contracts.

The raw data contains source code of two smart contracts that have been used in this research. One is for the storages connected to our system (pre-harvesting period) and the other one is used during distribution of agriproducts ((post-harvesting period)). Storage Contract is used in pre-harvesting period. The owner of the contract (deployer) and connected storages will have an Ethereum wallet address. Action allowed for every type of accounts are constrained by modifiers which are onlyOwner and onlyStorage. Four types of events are emitted by this contract during adding a seed or during any violation occurs. Manual and auto self-checks for environment variables (temperature, humidity, light exposure) is done with this contract. For any kind of violation, a trigger with violation type, value and condition is logged into the blockchain. Distribution Contract comes into play in post-harvesting period. The chain of actors will all be bought under automation by this contract. Owner, producer, distributor, wholesaler and retailers will operate through Ethereum wallet address. Data like product name, batch id, selling price, quantity, date and block timestamp will be saved for each and every step of product distribution through this contract. Each actor has certain access control through modifiers. Functions like initiateDisribution, startDistribution, startWholesell and retailSell is used to store data from each step of distribution. These data are also logged into the Blockchain by events.

Click here for additional data file.

Additional Information and Declarations

Competing Interests

Author Contributions

Data Availability

The authors declare that they have no competing interests.

Tahmid Hasan Pranto conceived and designed the experiments, performed the experiments, analyzed the data, performed the computation work, prepared figures and/or tables, authored or reviewed drafts of the paper, and approved the final draft.

Abdulla All Noman analyzed the data, prepared figures and/or tables, and approved the final draft.

Atik Mahmud performed the experiments, prepared figures and/or tables, and approved the final draft.

AKM Bahalul Haque conceived and designed the experiments, analyzed the data, authored or reviewed drafts of the paper, and approved the final draft.

The following information was supplied regarding data availability:

Raw data and code are available in the Supplemental Files.

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
