# Peer review of "Blockchain and smart contract for IoT enabled smart agriculture"

_PeerJ Computer Science, doi:10.7717/peerj-cs.407_

## Round 0.1 · original submission · Minor Revisions

The reviews are all positive, please revise the manuscript by addressing the comments from two reviewers that suggested minor suggestions and resubmit, Thanks.

·

Basic reporting

no comment

Experimental design

no comment

Validity of the findings

no comment

Additional comments

This is an excellent use case and road map on how to enable Ubiquitous computing in agriculture workloads now by integrating IoT devices and the data they can acquire and report upon. Placing this data on a blockchain creates immutability and adds value to workflow by creating trust in transactions and adding a layer of truth to the workflow. Great job!

·

Basic reporting

1) Most of the article is written in clear, professional English. Some minor grammatical corrections are needed as suggested in the annotated PDF attached to the review.

2) References are provided to prove that blockchain based smart contracts can be helpful in streamlining distribution and supply in agricultural sector

3) Professional article structure is followed. Figure 4 can be more understandable if order of steps is added.

4) Advantages and disadvantages of the proposed system are included in the paper which will be useful for future extensions or improvements of the proposed system.

Experimental design

1) Research question is well defined by setting the context where a solution is required. Relevant previous work in similar context and evidence for the advantage of using blockchain and smart contracts are also included

2) Good overview of blockchain and smart contracts and their usecases is included in the paper. Application of blockchain based smart contracts in agricultural sector is provided with reference to previous work in the area.

It will be good to answer the following questions:

3) Why MQTT was chosen for the design?

4) What is the source of cryptocurrency to the different type of actors in the system so that they can pay for the gas cost while submitting transactions to blockchain network?

Validity of the findings

1) It will be good to comment/document the code provided so that it is easy for reader to understand and use

2) How each batch is tracked in the supply and distribution chain can be useful to add in the design section? For example: barcodes or QR codes attached to the package? It will be good to answer the question.

3) It is mentioned that future goal is to deploy the system on their own blockchain. It will be good to mention the type of blockchain that will be used (Permissioned, public or hybrid). The type of blockchain can suggest how centralized or decentralized the whole system is.

Additional comments

Inclusion of IoT devices in the blockchain based agricultural food chain is a very interesting concept. It will be good to think about how these IoT devices can be authenticated , authorized and tracked in the system.

·

Basic reporting

Some small comments relating to a few references, and some of the language used, which could be improved in places...

“Contrarily, the farmers are the lowest paid seller in the whole chain. The price in retail shops is much higher, sometimes twice or thrice, than that the price sold by the farmers.”
Line 40, any reference here that can be cited?

Line 62 - missing word after mass

Eg line 80/81, Blockchain doesn’t need to be capitalised, it should be written “blockchain”

86 - should be “tamperproof”

(Nakamoto  2019) - is that the right reference?

140 - absurd not really the right word here - infeasible?

178 - By far, blockchain is the most secure way to deal with any kind of data. <⏤ I'm not sure about this claim

200 - usually would refer to author by last name, so Szabo not Nick here

211 - I don’t think Nakamoto 2008 is the correct reference here, for Ethereum

Various - check capitalisation of Remix and Solidity

495 - demonstration not demo (also other places)

Figure 10 - text is a bit small -

Experimental design

Seems good, meets criteria.

Validity of the findings

Seems good, well explained.

Additional comments

Thank you for your paper, which I enjoyed reading. It provides a good explanation of the situation being addressed, and the proposed solution. One thing that could be considered in the future is security/authenticity of the data from the IOT devices into the network. I particularly liked the very clear diagrams.

---

## Round 0.2 · accepted · Accept

Thanks for addressing the concerns.

·

Basic reporting

I note that the latest revision has changed "Ethereum" to "ethereum" which I don't think is correct - as a name, Ethereum should have capital "E", as in the original submission.

Experimental design

No further comments.

Validity of the findings

No further comments.

Additional comments

Thank you for making the suggested changes, the paper is looking very good now.

---

## Author Rebuttal · Round 0.2

Dear Editor,                                                December 17. 2020

We cordially thank the reviewers for reviewing our paper. The reviewers were good for the improvement of the paper.

We have tried to accommodate and address all the reviewers' comments to the best of our knowledge. The repository url is added at the conclusion and the code is commented in the github code repository.

After integrating all the comments, we believe it is now ready for PeerJ submission.

AKM Bahalul Haque
Lecturer, Dept. of Electrical and Computer Engineering
North South University.

On behalf of all the authors.

# Reviewer 1 : Tommy Cooksey

*Basic reporting*

*no comment*

*Experimental design*

*no comment*

*Validity of the findings*

*no comment*

*Comments for the author*

*This is an excellent use case and road map on how to enable Ubiquitous computing in agriculture workloads now by integrating IoT devices and the data they can acquire and report upon. Placing this data on a blockchain creates immutability and adds value to workflow by creating trust in transactions and adding a layer of truth to the workflow. Great job!*

**Author response:** As no changes were suggested, this review doesn't have any response from the authors.

# Reviewer 2 : Swapna Kr

*Basic reporting*

*1)Most of the article is written in clear, professional English. Some minor grammatical corrections are needed as suggested in the annotated PDF attached to the review.*

*2) References are provided to prove that blockchain based smart contracts can be helpful in streamlining distribution and supply in agricultural sector.*

*3) Professional article structure is followed. Figure 4 can be more understandable if order of steps is added.*

**Author response:** Thank you for this suggestion. We have updated figure 4 while adding the order and provided some more explanation in the manuscript about the steps. These changes are shown in line 348-354.

*4) Advantages and disadvantages of the proposed system are included in the paper which will be useful for future extensions or improvements of the proposed system.*

*Experimental design*

*1) Research question is well defined by setting the context where a solution is required. Relevant previous work in similar context and evidence for the advantage of using blockchain and smart contracts are also included.*

*2) Good overview of blockchain and smart contracts and their usecases is included in the paper. Application of blockchain based smart contracts in agricultural sector is provided with reference to previous work in the area.*

*It will be good to answer the following questions:*

*3) Why MQTT was chosen for the design?*

**Author response:** We have added the answer to this question in paragraph 2 of the system overview section. The main reason for choosing MQTT was to establish a server for all kinds of data that goes through our system. As blockchain has storage constraints, only sophisticated and

necessary data is being stored and tracked by blockchain. On the other hand, data, in general, is stored in MQTT servers, which are much more convenient for our purpose than standard HTTP. MQTT protocol requires the IoT devices to subscribe to a  topic once to send data, which in the case of HTTP would be needed to establish this connection each and every time the IoT devices try to send data. This change has also been added to the manuscript, which can be found in line 355-365.

*4) What is the source of cryptocurrency to the different type of actors in the system so that they can pay for the gas cost while submitting transactions to blockchain network?*

**Author response:** This cryptocurrency or ether is being used from their Ethereum wallet. In a real-world project, actors will have Ethereum wallet from where the transaction costs will be deducted. We used ethereum IDE, which comes with some dummy wallets for testing a system.

*Validity of the findings*

*1) It will be good to comment/document the code provided so that it is easy for reader to understand and use.*

**Author response:** Thank you for bringing this to our knowledge. Relevant comments have been added to all the codes.

*2) How each batch is tracked in the supply and distribution chain can be useful to add in the design section? For example: barcodes or QR codes attached to the package? It will be good to answer the question.*

**Author response:** Thank you for pointing this out. For demonstration purposes, we used batch codes to track items. Blockchain is well known for handling the online transaction. Showing the usability of blockchain for improving the agricultural sector is our main purpose of this research. So, we confined our focus on the scope of blockchain in the field of agriculture. Building a fully functional application is our next research interest. So, we took your advice sincerely and included this as one of our future research directions, which can be found in line 628.

*3) It is mentioned that future goal is to deploy the system on their own blockchain. It will be good to mention the type of blockchain that will be used (Permissioned, public or hybrid). The type of blockchain can suggest how centralized or decentralized the whole system is.*

**Response**: We sincerely thank you for mentioning this. We will use a permissioned blockchain, and this modification has been added to the manuscript in the conclusion section. This change can be found in line 626.

*Comments for the author*

*Inclusion of IoT devices in the blockchain based agricultural food chain is a very interesting concept. It will be good to think about how these IoT devices can be authenticated , authorized and tracked in the system.*

**Response**: We agree entirely with you. But, as this is falling outside our research target, which is essentially encircled to blockchain, we will try to incorporate your suggestion in our future researches.

# Reviewer: Iain Barclay

*Basic reporting*

*Some small comments relating to a few references, and some of the language used, which could be improved in places...*

**Response**: We agree with you, and we have incorporated your suggestions throughout the manuscript.

*"Contrarily, the farmers are the lowest paid seller in the whole chain. The price in retail shops is much higher, sometimes twice or thrice, than that the price sold by the farmers."*

*Line 40, any reference here that can be cited?*

**Response**: Thanks for pointing this out. A citation from the US Bureau of Labor Statistics (BLS) has been added to this statement, which can be found in lines 38-40.

*Line 62 - missing word after mass*

**Response**: Thanks for pointing to this. We have changed the word from "mass" to "the consumers," which can be found in line 64.

*Eg line 80/81, blockchain doesn't need to be capitalised, it should be written "blockchain"*

**Response**: We have incorporated your suggestions throughout the manuscript. All the capitalized "Blockchain" has been converted to "blockchain".

*86 - should be "tamperproof"*

**Response:** The correction has been made to the entire manuscript. These changes can be found in lines 136 and 234.

*(Nakamoto 2019) - is that the right reference?*

**Response**: This citation has been changed from "Nakamoto 2019" to "Nakamoto 2008". The reference has also been updated.

*140 - absurd not really the right word here - infeasible?*

**Response**: We agree with you. The word infeasible is the right word here. We corrected the word, which can be found in line 142.

*178 - By far, blockchain is the most secure way to deal with any kind of data. <□ I'm not sure about this claim*

**Response**: We agree with you. This claim was questionable. We changed our statement to, blockchain is one of the most secure ways of dealing with the electronic transaction. But, the usage of blockchain in real-world systems that deal with raw data has been shown by many researchers." This change can be found in line 180-182.

*200 - usually would refer to author by last name, so Szabo not Nick here.*

**Response**: We agree with you, and we've changed the reference from "Nick" to "Szabo." It can be found in line 203.

*211 - I don't think Nakamoto 2008 is the correct reference here, for Ethereum.*

**Response:** We agree that this is not the right citation and updated it. This update can be found in line 212-214.

*Various - check capitalisation of Remix and Solidity.*

**Response:** All the word has been capitalized to "Remix" and "Solidity."

*495 - demonstration not demo (also other places)*

**Response:** This suggestion has been followed throughout the whole manuscript. All "demo" has been replaced by "demonstration" throughout the paper.

*Figure 10 - text is a bit small –*

**Response**: This picture is a screenshot of Remix IDE's console output. We could not change the font size of the console output and tried our best by stretching the console to maximum size and then taking the snapshot. This was our best screenshot.

***Experimental design**script*

*Seems good, meets criteria.*

***Validity of the findings***

*Seems good, well explained.*

***Comments for the author***

*Thank you for your paper, which I enjoyed reading. It provides a good explanation of the situation being addressed, and the proposed solution. One thing that could be considered in the future is security/authenticity of the data from the IOT devices into the network. I particularly liked the very clear diagrams.*